# Risks and Benefits of SGLT-2 Inhibitors for Type 1 Diabetes Patients Using Automated Insulin Delivery Systems—A Literature Review

**DOI:** 10.3390/ijms25041972

**Published:** 2024-02-06

**Authors:** Viviana Elian, Violeta Popovici, Oana Karampelas, Gratiela Gradisteanu Pircalabioru, Gabriela Radulian, Madalina Musat

**Affiliations:** 1Department of Diabetes, Nutrition and Metabolic Diseases, “Carol Davila” University of Medicine and Pharmacy, 5-7 Ion Movila Street, 020475 Bucharest, Romania; viviana.elian@umfcd.ro (V.E.); gabriela.radulian@umfcd.ro (G.R.); 2Department of Diabetes, Nutrition and Metabolic Diseases, “N. C. Paulescu” National Institute of Diabetes, Nutrition and Metabolic Diseases, 020475 Bucharest, Romania; 3“Costin C. Kiriţescu” National Institute of Economic Research—Center for Mountain Economics (INCE-CEMONT) of Romanian Academy, 725700 Vatra-Dornei, Romania; 4Department of Pharmaceutical Technology and Biopharmacy, Faculty of Pharmacy, “Carol Davila” University of Medicine and Pharmacy, 6 Traian Vuia Street, 020945 Bucharest, Romania; oana.karampelas@umfcd.ro; 5eBio-Hub Research Centre, National University of Science and Technology Politehnica Bucharest, 061344 Bucharest, Romania; gratiela.gradisteanu@icub.unibuc.ro; 6Research Institute, University of Bucharest, 061344 Bucharest, Romania; 7Academy of Romanian Scientists, 54 Splaiul Independentei, 050094 Bucharest, Romania; 8Department of Endocrinology, “Carol Davila” University of Medicine and Pharmacy, 030167 Bucharest, Romania; 9Department of Endocrinology IV, “C. I. Parhon” National Institute of Endocrinology, 011863 Bucharest, Romania

**Keywords:** SGLT-2 inhibitors, Type 1 diabetes mellitus, automated insulin delivery systems, diabetic ketoacidosis

## Abstract

The primary treatment for autoimmune Diabetes Mellitus (Type 1 Diabetes Mellitus-T1DM) is insulin therapy. Unfortunately, a multitude of clinical cases has demonstrated that the use of insulin as a sole therapeutic intervention fails to address all issues comprehensively. Therefore, non-insulin adjunct treatment has been investigated and shown successful results in clinical trials. Various hypoglycemia-inducing drugs such as Metformin, glucagon-like peptide 1 (GLP-1) receptor agonists, dipeptidyl peptidase-4 (DPP-4) inhibitors, amylin analogs, and Sodium-Glucose Cotransporters 2 (SGLT-2) inhibitors, developed good outcomes in patients with T1DM. Currently, SGLT-2 inhibitors have remarkably improved the treatment of patients with diabetes by preventing cardiovascular events, heart failure hospitalization, and progression of renal disease. However, their pharmacological potential has not been explored enough. Thus, the substantial interest in SGLT-2 inhibitors (SGLT-2is) underlines the present review. It begins with an overview of carrier-mediated cellular glucose uptake, evidencing the insulin-independent transport system contribution to glucose homeostasis and the essential roles of Sodium-Glucose Cotransporters 1 and 2. Then, the pharmacological properties of SGLT-2is are detailed, leading to potential applications in treating T1DM patients with automated insulin delivery (AID) systems. Results from several studies demonstrated improvements in glycemic control, an increase in Time in Range (TIR), a decrease in glycemic variability, reduced daily insulin requirements without increasing hyperglycemic events, and benefits in weight management. However, these advantages are counterbalanced by increased risks, particularly concerning Diabetic Ketoacidosis (DKA). Several clinical trials reported a higher incidence of DKA when patients with T1DM received SGLT-2 inhibitors such as Sotagliflozin and Empagliflozin. On the other hand, patients with T1DM and a body mass index (BMI) of ≥27 kg/m^2^ treated with Dapagliflozin showed similar reduction in hyperglycemia and body weight and insignificantly increased DKA incidence compared to the overall trial population. Additional multicenter and randomized studies are required to establish safer and more effective long-term strategies based on patient selection, education, and continuous ketone body monitoring for optimal integration of SGLT-2 inhibitors into T1DM therapeutic protocol.

## 1. Introduction

Glucose is an essential nutrient for the human body; it represents the principal energy source for live cells, depending on the bloodstream for a continuous supply. Glucose cellular uptake is essential for metabolism, growth, and homeostasis. Having a polar nature and high molecular mass, glucose cannot penetrate the lipid cell membrane by simple diffusion. It needs a carrier-mediated process, occurring by passive and active transport mechanisms. The carriers are proteins that differ by substrate specificity, distribution, and regulatory mechanisms. Uniporters are sodium-independent, transporting glucose across the plasma membrane by facilitated diffusion (GLUTs and SWEETs) [1].

GLUTs are proteins containing 12 membrane-spanning regions with amino and carboxyl terminals inside the cells. Three subclasses of facilitative transporters have been identified based on this and multiple sequence alignment studies. Class I contains GLUT1-4, and class II includes the other four members (GLUT5, GLUT7, GLUT9, and GLUT11). Finally, GLUT6, GLUT8, GLUT10, GLUT12, and GLUT13 belong to class III. Insulin regulates blood glucose levels by stimulating physiological responses in target tissues. The glucose transport proteins (GLUT1 and GLUT4) facilitate glucose transport into insulin-sensitive cells. GLUT1 is insulin-independent and is widely distributed in different tissues. GLUT4 is insulin-dependent and responsible for most glucose transport into muscle and adipose cells in anabolic conditions. In skeletal muscle and adipose tissue, insulin promotes membrane trafficking of the glucose transporter GLUT4 from storage vesicles to the plasma membrane, thereby facilitating glucose uptake from the circulation.

SWEETs are hepta-helical proteins composed of a tandem repeat of three transmembrane domains (TMs) connected by a linker-inversion. Human SWEET1 comprises 221 amino acids.

Symporters are responsible for active glucose transport against its concentration gradient; they are sodium-dependent glucose cotransporters (SGLTs) [1]. They complement pancreatic hormones (insulin and glucagon) in maintaining sugar homeostasis through blood glucose level regulation. All SGLTs are 60- to 80-kDa proteins containing 580–718 amino acids. They have 14 transmembrane helices, of which the -COOH and -NH2 terminals face the extracellular space. The most important data regarding the location, substrate, and roles of glucose transporters are displayed in Table 1.

Twelve members of the SGLT family are identified in the human genome; Table 1 contains the six most known. The first SGLTs (SGLT-1 and 2) are pivotal in maintaining normoglycemia in healthy humans [2]. SGLT-1 is expressed in the small intestine’s epithelial cells (enterocytes), ensuring over 80% glucose absorption [3]. It involves glucose transfer from the enterocytes into the intracellular space by facilitated diffusion and then glucose release into the blood circulation. SGLT-1 could also be found in other organs [4,5]: in the kidneys, uterus, prostate, lungs, tongue, pancreas, liver, salivary glands, eyes, heart, brain, and skeletal muscles [5,6,7,8,9,10].

**Table 1 ijms-25-01972-t001:** The properties of the most known types of glucose transporters.

Transporter	Location	Substrate	Function
1. Passive transport, sodium-independent transporters [11]			
1.1. GLUTs			
GLUT1 [11,12,13]	Ubiquitous distribution: Kidney, colon, retina, placenta, myocardium, adipose tissue, brain, blood–brain barrier, blood–tissue barrier, and many fetal tissues	D-glucose, D-galactose, D-glucosamine, Glucose analogs: 2-deoxy-D-glucose and 3-O-methyl-D-glucose	Glucose and other carbohydrates’ uptake in cells. Primarily responsible for the transport of glucose across the blood–brain barrier. Its expression is age-related.
GLUT2 [14,15]	Liver, intestine, kidney, pancreatic beta-cells, muscle, fat, heart central nervous system (neurons, astrocytes, and tanycytes)	Glucose Galactose, Fructose, Mannose, Glucosamine	Glucose absorption in the intestine and kidney. Glucose sensor in beta cells and in Nervous system.
GLUT3 [16]	Brain	Glucose	Neuronal glucose absorption.
GLUT4 [17]	Insulin-sensitive tissues of adipose, heart, and skeletal muscle	Glucose	Insulin-dependent glucose uptake.
GLUT5 [18]	Small intestine, kidney, and sperm. Fat and skeletal muscle (lower levels)	Fructose	Intestinal fructose absorbtion. In adipose and muscle, GLUT5-mediated fructose transport is responsive to insulin stimulation.
GLUT6 [19]	Lysosomal membranes of lymphocytes and spleen and brain cells	Glucose, Fructose	Glycolysis modulator in inflammatory macrophages. It has a low affinity for both saccharides.
GLUT7 [20]	Small intestine, colon, testis, and prostate	Glucose, Fructose	Intestinal transport of fructose and glucose. Low capacity.
GLUT8 [17,21]	Testis, spermatozoa, lactating mammary gland alveolar cells	Glucose, Fructose, Galactose, Trehalose	Glucose, fructose, trehalose, and galactose transporter. High affinity for glucose.
GLUT9 [22]	Liver, kidney, placenta	Uric acid	Renal urate reabsorption. It plays an essential role in the placental uric acid transport system.
GLUT10 [23]	Arterial smooth muscle cells (ASMC), liver, and endocrine pancreas	Dehydroascorbic acid	Maintain the integrity of significant arteries by regulating redox homeostasis and mitochondrial function. The targeting of GLUT10 to mitochondria is increased in ASMCs under stress and aging conditions, which enhances dehydroascorbic acid uptake and maintains intracellular ascorbic acid levels.
GLUT11 [17]	Heart, kidney, skeletal muscle, adipose tissue, and pancreas	Fructose Glucose	Its sequence has the highest similarity to GLUT5, suggesting that fructose may be the preferred substrate for transport.
GLUT12 [24]	Adipose, heart and skeletal muscle cells, and mammary gland alveolar cells	Glucose	Insulin-independent glucose transporter in the heart.
GLUT13 [25]	Brain, high abundance in the hippocampus, hypothalamus, cerebellum, and brainstem	Myo-inositol	Proton-coupled Myo-inositol transport in the brain.
GLUT14 [26]	Testis, utero-placental tissues	Glucose	Supply glucose for the establishment and maintenance of pregnancy.
1.2. SWEETs			
SWEET1	Oviduct, epididymis, intestine, and pancreatic beta cells.	Monosaccharides, Disaccharides	Transport various mono- and disaccharides. Mediate both cellular uptake and efflux. Low affinity. Human SWEET1 does not promote glucose uptake but could mediate a weak efflux.
2. Active transport, Sodium-dependent transporters [27]			
SGLTs			
SGLT-1	Apical membranes of small intestinal cells	Glucose	Absorption of glucose from intestinal content. High affinity.
	Straight cells (S3 cells) of the proximal tubule of the nephron	Glucose	Reabsorption of remaining glucose from urine filtrate. High affinity. Low capacity.
SGLT-2	Proximal convoluted tubule of nephron (S1 and S2 cells)	Glucose	Reabsorption of bulk plasma glucose from glomerular filtrate. Low affinity. High capacity.
SGLT-3	Intestine, testes, uterus, lung, brain, thyroid	Glucose	Function as a glucose sensor for controlling glucose levels in the gut and brain.
SGLT-4	Intestine, kidney, liver, brain, lung, uterus, pancreas	Mannose, Fructose, Glucose	Absorption and/or reabsorption of mannose, 1,5-anhydro D-glucitol, fructose, and glucose. Low affinity.
SGLT-5	Kidney cortex	Glucose, Galactose	Transport of glucose and Galactose.
SGLT-6	Brain, kidney, intestine	Myo-inositol, D-chiro-inositol, Glucose	D-chiro-inositol is the main substrate. High affinity for Myo-inositol and low affinity for glucose.

In the kidneys, glucose is reabsorbed along the proximal tubules after glomerular filtration. SGLT-2 and SGLT-1 are expressed in the luminal membrane of the early (S1 and S2 segment) and late proximal tubule (S2 and S3 segment), respectively. They reabsorb around 97% (SGLT-2) and 3% (SGLT-1) of filtered glucose in normoglycemic conditions. A significant capacity of SGLT-1 to reabsorb glucose is unmasked by SGLT-2 inhibition (~40–50% reabsorption in normoglycemia) and during hyperglycemia, which both enhance glucose delivery to the late proximal tubule. Consequently, diabetes-induced hyperglycemia or SGLT-2 inhibition increases the SGLT-1 inhibition-induced increase in glucose excretion, which provides the renal rationale for dual SGLT-1/2 inhibition. Both SGLT-1 and 2 display high concentrations in Diabetes Mellitus (DM), contributing to a significant rise in blood glucose (BG) levels. Therefore, a basic concept of the therapeutic strategy is to inhibit both SGLT-1 and 2, avoiding the glucose burden by inhibiting the uptake of dietary glucose in the intestine or excreting the glucose filtered by the kidneys into the urine.

Currently, the SGLT-2 inhibitors (SGLT-2is, gliflozins, [28]) have remarkable applications in DM therapy [29]. They are a relatively new class of oral hypoglycemia-inducing agents in continuous expansion that block renal glucose reabsorption [30]. SGLT-2 inhibitors approved by the United States Food and Drug Administration (FDA) and European Medicinal Agency (EMA) include Empagliflozin, Dapagliflozin, Canagliflozin, Ertugliflozin, Ipragliflozin, Tofogliflozin, Luseogliflozin, and Remogliflozin. In addition to decreasing blood glucose levels, SGLT-2 inhibitors can reduce body weight, normalize lipid profile and serum uric acid levels, diminish visceral adiposity, and lower blood pressure [31]. Notably, recent cardiovascular outcome trials (CVOTs) assessing SGLT-2 inhibitors have shown improvements in renal and cardiovascular outcomes in patients with and without Type 2 Diabetes Mellitus (T2DM). Based on accumulating evidence, the American Diabetes Association (ADA)/European Association of the Study for Diabetes (EASD) recommends SGLT-2 inhibitors as elective therapy for T2DM [32]. They are also used as an adjunct treatment in T1DM, in combination with insulin in adults with a body mass index (BMI) of ≥27 kg/m^2^, inadequately controlled with insulin only. In patients affected by neoplastic processes, these proteins are also over-expressed in various tumor cells [33,34]. Therefore, sodium-glucose cotransporters represent new targets in multiple pathologies, the research interest in their properties continuously increases, and the scientific database is constantly updated with new and significant reports. Thus, the present study displays the most recent data about the risks and benefits of gliflozins in patients with Type 1 Diabetes Mellitus (T1DM) with automated insulin delivery (AID) systems. Two databases (Google Scholar and PubMed) and Mendeley Cite v1.65.1/2024 were used for open-access articles/reviews published between 2014 and 2024. The keywords used were SGLT-2 inhibitors, Type 1 Diabetes Mellitus, automated insulin delivery systems, and diabetic ketoacidosis. The last search was performed on 25 January 2024.

## 2. SGLT-2 Inhibitors in DM Therapy

Due to considerable glycemic and non-glycemic benefits, the FDA approved several selective SGLT-2 inhibitors for T2DM treatment [35]. Currently, all available gliflozins as drugs commercialized in pharmacies (Canagliflozin, Dapagliflozin, Empagliflozin, Ertugliflozin, and Bexagliflozin) have T2DM as a main common indication. Heart failure (HF) is mentioned for Dapagliflozin and Empagliflozin, and chronic kidney disease (CKD) only for Dapagliflozin.

Sotagliflozin, a dual SGLT-1/2 inhibitor, is the only representative officially indicated in T1DM.

Other new similar compounds (Ipragliflozin, Luseogliflozin, Tofogliflozin, and Remogliflozin) are investigated in clinical studies. The most recent SGLT inhibitors are investigated using pharmacokinetic and pharmacodynamic analyses (Remoglicoflozin [36,37,38,39], Henaglicoflozin [40,41,42,43], and Licoglicoflozin [44,45,46,47]).

All data are summarized in Table 2.
ijms-25-01972-t002_Table 2Table 2FDA-approved/in-trials SGLT-2 Inhibitors for T2DM therapy.SGLT-2InhibitorActiveIngredient(s)Daily Dose(mg)Brand NameCompanySelective SGLT-2 InhibitorsCanagliflozin[48,49,50]Canagliflozin100InvokanaJanssen-CilagInternational NVBeerse, BelgiumCanagliflozin + Metformin50/50050/1000150/500150/1000InvokametCanagliflozin + Metformin extended-releaseInvokamet XRDapagliflozin[51,52,53,54,55]Dapagliflozin510ForxigaAstraZeneca ABSödertäljeSwedenDapagliflozin + Metforminextended-release5/10005/850Xigduo XRDapagliflozin + Saxagliptin5/10QternEmpagliflozin[56,57,58,59]Empagliflozin1025JardianceBoehringer Ingelheim International GmbHIngelheim am Rhein,GermanyEmpagliflozin + Linagliptin10/525/5GlyxambiEmpagliflozin + Metformin5/10005/85012.5/100012.5/850SynjardyEmpagliflozin + Metforminextended-release25/1000Synjardy XRErtugliflozin[60,61]Ertugliflozin515SteglatroMerck Sharp & DohmeHaarlem,NetherlandErtugliflozin + Metformin2.5/8507.5/850SeglurometErtugliflozin + Sitagliptin5/10015/100SteglujanBexagliflozin[62,63,64,65,66,67]Bexagliflozin20BrenzavvyTheracosBio, LLCMarlborough, MA, USAIpragliflozin[68,69,70,71,72]Ipragliflozin2550SuglatAstellas Pharma LTD,Addlestone, UKLuseogliflozin[73,74,75,76,77,78]Luseogliflozin2.55LusefiTaisho Pharmaceutical Holdings Co., Ltd.,Tokyo, JapanTofogliflozin[79,80,81]Tofogliflozin2040AplewayChugai Pharmaceutical Co., Ltd.,Tokyo, JapanRemogliflozin[37,38,39,82,83]Remogliflozin100RemogliflozinetabonateGlaxoSmithKline plc, Brentford, UKGlenmark Pharmceuticals Ltd.,Mumbai, IndiaHenagliflozin[40,41,42,84,85]Henagliflozin510SHR3824Jiangsu Hengrui Pharmaceuticals Co., Ltd., Lianyungang,ChinaDual SGLT-2 + SGLT-1 InhibitorsSotagliflozin[86,87,88,89,90,91,92,93,94]Sotagliflozin200400ZynquistaInpefaLexiconPharmaceuticals, Inc.,The Woodlands, TX, USALicogliflozin[44,45,46,47,95]LicogliflozinNo data LIK-066Novartis AGBasel, Switzerland

### 2.1. Benefits of SGLT-2 Inhibitors

Globally, SGLT-2 inhibitors are approximately the most prescribed oral antidiabetic drugs. Their beneficial effects beyond glycemic control include weight loss, protection against major cardiovascular events, blood pressure reduction, and delaying the progression of chronic kidney disease (CKD).

#### 2.1.1. Weight Loss

SGLT-2i directly reduces body weight by removing glucose through urine, thus increasing calorie loss. Glycosuria due to the SGLT-2 inhibitors leads to lower plasma glucose and insulin levels, followed by increased fasting and post-meal glucagon concentration. While blood glucose concentration is diminished, lipid storage is mobilized to be used as an energy substrate. The persistent excretion of glucose in urine induces increasing gluconeogenesis, suppresses tissue glucose disposal and glucose oxidation, accelerates lipolysis and fat oxidation, and enhances ketogenesis. The overall result of these metabolic changes resembles a fasting state, which will cause the loss of fat mass and weight in the long run [96]. Another study shows the concomitance of weight loss and Hemoglobin A1c level diminution during the therapy with SGLT-2is [97]. Thus, sodium-glucose cotransporter inhibitors could be called mimic-fasting medication to prevent cardiovascular complications [98,99]. SGLT-2 inhibitors reduce body weight by around 2–4 kg; this average is constant for all representatives and persists for up to 4 years [96].

#### 2.1.2. Heart Protection

Renal function improvement and preservation through hemodynamic and nonhemodynamic mechanisms are essential for SGLT-2 inhibitors in heart protection [100]. It is necessary to know the interactions between all biochemical processes, the chronology of all changes, and their correlation with the cardiovascular benefits of gliflozin therapy [101]. The potential mechanisms [31] are diuresis and natriuresis, changes in myocardial energetics, increased erythropoietin (EPO) production and erythropoiesis, changes in reno-cardiac signaling, inhibition of the sympathetic nervous system, inhibition of the Na^+^/H^+^ Exchanger-1 (NHE1), inhibition of the NOD-like receptor family, pyrin domain-containing 3 (NLRP3) inflammasome, potential vascular effects, and reducing uric acid levels in serum [102].

#### 2.1.3. Kidney Protection

The highlight of SGLT-2i kidney protection led to significant interest in gliflozins’ broader applications in CKD therapy [103]. The potential mechanisms of their protective effects could be glucosuria, inducing natriuresis and osmotic diuresis and leading to reduced plasma volume and lower blood pressure, reducing proteinuria, and delaying CKD progression in patients with albuminuria [104], hemodynamic changes at the systemic and glomerular levels, metabolic pathway, and decreasing oxidative stress and inflammation.

### 2.2. Adverse Effects of SGLT-2 Inhibitors

#### 2.2.1. Acute Kidney Injury (AKI)

The SGLT-2is induces glucose and sodium overexcretion, conducting osmotic diuresis. It may lead to hyperosmolarity and dehydration, increasing the risk of AKI [105]. Recent studies rigorously demonstrate that SGLT-2 inhibitors are safe for the kidneys and do not predispose to AKI [106].

#### 2.2.2. Polyuria

Significant glucosuria and natriuria induced by SGLT-2is lead to polyuria due to osmotic diuresis.

#### 2.2.3. Euglycemic Diabetic Ketoacidosis (DKA)

Production of ketone bodies may show a dual face. The heart and brain can rapidly use them as energetic substrates, avoiding the accumulation of fatty acid or glucose metabolites. Normal ketone levels are <0.6 mmol/L. Most patients treated with SGLT-2is concomitantly have ketosis (ketone levels are slightly increased 0.6–1.5 mmol/L), and they do not develop ketoacidosis. Thus, the tendency toward ketosis can be augmented by a low-carb intake in patients who are trying to lose weight. Ketone bodies are produced by the oxidation of fatty acids in the liver as a source of alternative energy, generally occurring in glucose-limiting conditions. Elevated blood levels of acetoacetate (AA), 3-β-hydroxybutyrate (BHB), and acetone are known as hyperketonemia. BHB serum concentration increases after fasting but should not exceed 0.4 mmol/L. High levels of circulating ketones are linked to oxidative stress and numerous morbid conditions. 

The risk of DKA is 1.5–3 mmol/L. DKA manifests when the ketone level is highly increased, over 3 mmol/L. Possible mechanisms [105] of SGLT-2 inhibitors associated with euglycemic DKA could be noninsulin-dependent glucose clearance, hyperglucagonemia, and volume depletion.

#### 2.2.4. Genito-Urinary Tract Infections

Glycosuria is the main factor implied in these infectious diseases, being a favorable medium for the growth of bacterial and fungal strains. They are more frequent in females and can be easily treated with non-expensive drugs.

#### 2.2.5. Bone Fractures and Amputation Risk

Potential mechanisms for fractures [107] could be volume contraction leading to dizziness and falls, possible effects on calcium, phosphate, and vitamin D homeostasis, and reduction in bone mineral density. The amputation risk [108] is linked to peripheral vascular disease, neuropathy, history of diabetic foot ulcer, and previous history of amputations. Euglycemic DKA and its potential association with a significant risk for lower-extremity amputation represent sporadic but possible fatal adverse events of SGLT-2is. Some studies reported that SGLT-2 inhibitors in T2DM may cause latent autoimmune DM of adulthood (LADA) [109].

### 2.3. SGLT-2 Inhibitors in the Therapy of DM Complications and Comorbidities

Despite the common knowledge that DM is associated with most known complications (traditional complications such as stroke, coronary heart disease, and heart failure, peripheral neuropathy, retinopathy, diabetic kidney disease), an increased prevalence of cardiovascular pathology and a group of lesser-studied ones have been reported (cancer, infections, functional and cognitive disability, liver disease and affective disorders) [110].

The benefits of SGLT-2 inhibitors in DM complications and comorbidities could be explained through the dual redox behavior of gliflozins (Figure 1) because oxidative stress has an essential role in their onset and harmful evolution. High blood glucose levels induce ROS production, leading to overexpression of the SGLT-2 in tubular cells, exacerbating oxidative stress. SGLT-2is has demonstrated clear cardiovascular and renal protection due to its antioxidant properties. Severe systemic comorbidity that involves the whole body, leading to functional decline and harmful outcomes [111,112,113] is frailty [114,115,116,117,118,119,120] or functional disability [121,122,123,124], and its management is still debated [119,120]. SGLT-2i positively acts on cardiovascular complications, especially on the HF rehospitalization rate [125], during several potential mechanisms: improving cardiovascular energetics, reducing vascular tone and blood pressure, decreasing systemic inflammation, atheroprotective effects, reduction of vascular damage, and direct neuroprotective mechanisms (acetylcholinesterase inhibition and increase in cerebral levels of brain-derived neurotrophic factor). All benefits could be maintained through rigorous balancing between oxidant and antioxidant processes [126].

The anticancer potential of gliflozins depends on the blood glucose-lowering capacity [128,129]. SGLT-2 inhibitors induce apoptosis and DNA damage, reducing cancer cell proliferation [130,131,132,133,134,135,136,137] through mitochondrial membrane instability metabolic changes (oxidative phosphorylation, DNA synthesis, glycolysis, ATP and fatty acids level diminution, beta-oxidation, and ketone amount augmentation).

DM is also associated with cancer development through a complex mechanism. Treatment with SGLT-2 inhibitors can diminish the risk of cancer incidence in DM patients [138] because gliflozins have anticancer activity through various mechanisms, as previously shown [139,140,141,142,143,144,145]. Moreover, SGLT-2 inhibitors can protect DM patients against the cardiotoxic action of anticancer drugs [31].

### 2.4. DM Patient Adherence to SGLT-2is Therapy

The most common reasons for discontinuing treatment with SGLT-2is are frequent urination, genital infection, improved glycemic control, and renal dysfunction. A recent retrospective study [146] did not find a correlation between patients’ compliance and DM type, duration, diabetic control, renal function, or DM complications of diabetes in both groups. Only the age was correlated (the adherence negatively correlates with the patient’s age).

## 3. SGLT-2 Inhibitors in Therapy of T1DM Patients Using AID Systems

Despite the limited approval for SGLT-2is use in T1DM, this class became attractive for treating patients with T1DM in conjunction with insulin therapy. Therefore, SGLT-2 inhibitors may help patients achieve their Glycated hemoglobin (HbA1c) goals by decreasing the insulin requirements without inducing hypoglycemic episodes and weight gain.

Multiple trials and meta-analyses assessed SGLT-2is as an add-on therapy to insulin treatment in patients with T1DM; however, their results are still controversial [147,148,149].

First published data from randomized trials using SGLT-2i in the treatment of patients with T1DM came from Canagliflozin and proved beneficial in metabolic control with a decrease in HbA1c and total insulin daily dose with no increased risk of hypoglycemia and decrease in body weight [48,150,151,152,153]. Data on Dapagliflozin reported as DEPICT-1 and 2 studies were published [154], followed by EASE trials with data on Empagliflozin [155]. Gliflozins were beneficial in HbA1c, mean glucose, TIR, insulin total daily dose (TDD), weight, and systolic blood pressure. Unfortunately, results from several meta-analyses on the randomized studies showed an increase in the frequency of DKA (RR = 2.81–5.04), urinary tract infections, and genital infections in SGLT-2is compared to placebo [57,59,155,156,157,158].

Sotagliflozin, the dual inhibitor of SGLT-2 and SGLT-1, also proved beneficial for glycemic control from inTANDEM randomized controlled trials with less frequent adverse events [159,160,161,162]. The results showed sustained HbA1c and TIR reduction, weight loss, lower total daily insulin dose, and less severe hypoglycemia [87,90,94,163,164,165,166]. Sotagliflozin is the only gliflozin approved for T1DM as an adjunct to insulin therapy in the EU and Japan [167,168,169].

Management of T1DM involves intensive insulin therapy. Multiple insulin injections (MDI) are now replaced by continuous subcutaneous insulin infusion (CSII) systems, also known as insulin pumps. The advantages of using insulin pumps are multiple, starting from better glycemic control through more efficient adjustments of the amount of insulin delivered, decreasing the frequency of hypoglycemia, and increasing the quality of life of T1D patients. The association of continuous glycemic monitoring systems (CGMS) with CSII treatment significantly improved glycemic control, and the CSII—CGMS connection was the first step toward automatic insulin delivery systems. The development of mathematical algorithms for AID systems, Model Predictive Control (MPC), Proportional-Integral-Derivative (PID), and Fuzzy Logic made it possible to automatically deliver insulin in response to continuously monitored interstitial fluid glucose levels.

Automated insulin delivery methods are increasingly common in patients with T1DM, with good results in improving glycemic control and quality of life [145]. The use of AID systems resulted in increased Time in Range (TIR), decreased Time below Range (TBR), reduced frequency of severe hypoglycemia, and glycemic variability [146,147,148,149,150]. However, AID systems have several limitations, including the indispensable self-counting of the mealtime carbohydrates for entering the system, announcing physical exercise, and regularly replacing the CGMS and pump components; also, if the system becomes unavailable, the patient must control the extreme blood glucose (BG) levels [145]. For optimal use, it is important to regularly verify the devices’ functionality and the software settings to ensure a suitable amount of insulin infusion and CGMS accuracy.

We have identified five randomized placebo-controlled trials that aimed to prove the benefits of SGLT-2 inhibitors as add-ons for patients with T1DM on closed-loop insulin therapy.

The DAPADream Study included adolescents and young adults who started on a fully automated closed-loop insulin delivery system that received 10 mg of Dapagliflozin at 12 h intervals and standardized mixed meals. For 24 h, CGMS data and biological samples were collected. The study demonstrated a significant increase in TIR (70–180 mg/dL) in the Dapagliflozin group compared to placebo (68 ± 6% vs. 50 ± 13%; *p* < 0.001), without an increase in time below 70 mg/dL (3.3 ± 6.0% vs. 3.1 ± 5.2%; *p* = 0.75) and a significant lower nocturnal glycemia. Dapagliflozin treatment resulted in a substantial reduction (on average by 22%) in the total daily dose of insulin by reducing the number of automated daily correction boluses by 24% and decreasing the total bolus dose by 38% (*p* < 0.001). Ketone levels slightly increased but remained in the normal range (<0.6 mmol/L), and urinary glucose excretion increased three-fold in the intervention group [170].

Empagliflozin was used in different studies and several formulations. Haidar et al. [158] used the 25 mg dose, while Garcia-Tirado [171] and Pasqua [156] used smaller doses (5 mg and 2.5 mg) to prevent adverse events such as DKA.

An open-label, crossover, non-inferiority trial aimed to assess whether 25 mg of Empagliflozin added to AID systems could reduce the need for carb-counting. The study enrolled 30 adults with T1DM randomized on two arms with Empagliflozin or placebo and three random sequences of prandial insulin strategy days: carbohydrate counting, simple and no meal announcement. Analysis showed that Empagliflozin added to AID allows replacing carb-counting with a simple meal announcement with TIR of 68 ± 16% vs. 70 ± 23%, similar TAR (30 ± 18% and 28 ± 23%), hypoglycemia (0%), and coefficient of variation (CV): 31 ± 7% vs. 30 ± 9%. In this study, carb-counting with Empagliflozin added remained the best solution as TIR was 84 ± 11% for this combination. Concerning adverse events, mean fasting ketones were higher with Empagliflozin vs. placebo (0.22 ± 0.18 vs. 0.13 ± 0.11 mmol/L; *p* < 0.001) [158].

In another study, 35 participants were randomized on two arms: 5 mg Empagliflozin and placebo. For each arm, participants were randomly assigned to a hybrid AID system Control-IQ (CIQ) followed by a predictive low-glucose suspension Basic-IQ system (BIQ) or BIQ followed by CIQ. Using 5 mg Empagliflozin and both CIQ or BIQ generated a significant increase in daily TIR vs. placebo without increasing the time spent in hypoglycemia. There was no significant difference between CIQ and BIQ when Empagliflozin was added (81% vs. 80%) regarding daily TIR. CIQ remained superior to BIQ overnight regarding higher percent TIR, with less time spent on hypoglycemia and hyperglycemia and lower glucose variability. The total daily insulin delivered by CIQ with Empagliflozin was lower than placebo during 24/7 and overnight periods, with insulin reductions of 24% and 35.3%, respectively. As one episode of DKA occurred and ketones were more frequently present in Empagliflozin patients, the authors suggest the possibility of using 2.5 mg of Empagliflozin in further studies to increase safety; still, the clinical benefit remains to be proven [170].

In a recently published study, a double-blind crossover randomized controlled trial during which the participants underwent three 14-day interventions of HCL therapy with the add-on of placebo, 2.5 mg Empagliflozin daily, or 5 mg Empagliflozin daily. The CGMS data were collected from the last 10 days of each intervention. Hence, the mean TIR (70–180 mg/dL) in the group with placebo was obtained at 59.0 ± 9.0% attendants. The corresponding values for the patients treated with 2.5 mg and 5 mg Empagliflozin were 71.6 ± 9.7% and 70.2 ± 8.0%, respectively. A higher proportion of participants achieved TIR > 70% when using Empagliflozin: 5% (15 of 24) and 45.8% (11 of 24) on 2.5 and 5 mg Empagliflozin, respectively, compared to 8.3% (2 of 24) on placebo. Empagliflozin also reduced the mean glucose level and total daily insulin dose. Events of ketosis < 0.6 mmol/L were few, with slightly more in the group with 5 mg Empagliflozin. Ketone levels > 1.5 mmol/L were rare (1 on placebo and 3 on 2.5 mg Empagliflozin), and all were related to catheter malfunction. The most reported adverse events were polyuria, thirst, and nausea [156].

A randomized, placebo-controlled crossover trial to prove the efficacy and safety of 25 mf of Empagliflozin in T1DM was performed [158], including 27 patients randomized on two groups: 25 mg Empagliflozin and placebo. Participants were randomly assigned to a closed-loop (CL) system followed by a sensor-augmented pump (SAP), or SAP followed by CL. Empagliflozin improved Time in Range with SAP therapy compared to placebo by 11.4% (7.7–15%, *p* < 0.0001), corresponding to 2.7 h (1.8–3.6 h) and with closed-loop therapy compared with placebo by 7.2% (3.5–10.9%, *p* < 0.0001)—1.7 h (0.8–2.6 h). The average daily insulin dose was reduced when Empagliflozin was used both on CL and SAP: 54.2 ± 28.0 Units per day (U/d) with closed-loop therapy plus Empagliflozin, 59.8 ± 31.2 U/d with closed-loop therapy plus placebo, 54.2 ± 28.0 U/d with SAP therapy plus Empagliflozin and 9.2 ± 29.8 U/d with SAP therapy plus placebo. Empagliflozin significantly increased the mean fasting ketone level with CL from 0.12 ± 0.06 mmol/L to 0.27 ± 0.15 mmol/L and with SAP from 0.11 ± 0.08 mmol/L to 0.17 ± 0.08 mmol/L. The authors suggest a synergistic effect of CL and SGLT-2i on ketone levels. They also found a negative correlation between BMI and ketone levels [172]. The data previously described are summarized in Table 3. The concern for DKA in patients with T1DM treated with SGLT-2i was addressed by the International Consensus on Risk Management of Diabetic Ketoacidosis in Patients with Type 1 Diabetes Treated WithSodium–Glucose Cotransporter (SGLT) Inhibitors [71] which recommended an appropriate patient selection for this therapy, continuous ketone monitoring, insulin dose adjustments with caution in reducing daily doses, or discontinuation of SGLT-2i if ketones persist or are in high levels [173,174].

Some brief reports of clinical cases provide data on potential DKA risk associated with SGLT-2i use in patients diagnosed with T1DM [175,176]. One of them is the report of Visser et al. [177], which describes the case of a 23-year-old woman with T1DM who developed DKA two weeks after the initiation of SGLT-2i therapy. She was obese, using an advanced hybrid closed loop (AHCL) therapy system (Medtronic 780G); Empagliflozin (12.5 mg daily) was prescribed to decrease her weight. After starting SGLT-2i treatment, the patient experienced a 49% reduction in total daily insulin dose by the AHCL system, particularly in auto basal and autocorrection and meal bolus doses, contributing to DKA’s development. The authors recommend caution in using SGLT-2is in T1DM patients on AHCL systems due to the increased risk of DKA. They suggest that a clear understanding of the effects of SGLT-2is on AHCL algorithms is essential before considering such treatment combinations. Therefore, specific recommendations, careful patient selection, and education on self-monitoring of blood ketones while using SGLT-2is are advised to prevent DKA development [178].

Another case report [179] involved a 54-year-old woman with a 15-year history of T1DM. She exhibited DKA symptoms after initiating an HCL insulin delivery system (Medtronic MiniMed 670G) while using an SGLT-2 inhibitor (Empagliflozin) as an add-on to insulin therapy. Several factors contributed to the development of DKA in this case, including canula malfunction, reduced basal insulin delivery by the automated system, and the effects of SGLT-2i. The authors highlighted the ‘automation complacency’ phenomenon, where patients become less vigilant in monitoring their condition due to reliance on automated systems. Genital and urinary tract infections were more prevalent in the case of SGLT-2i added to AID than placebo. Specific treatment and temporary or long-term SGLT-2i suspending contributed to the case’s resolution. Both cases are summarized in Table 4. 

**Table 3 ijms-25-01972-t003:** A summary of studies assessing the use of SGLT-2is for patients with T1DM treated by AID systems.

Study	SGLT-2i Used	Participants/ Duration	Aim	TIR	Hypo- Glycemia	Insulin Use	DKA
Biester et al., 2021 [180]	Dapagliflozin 10 mg (b.i.d.) vs. placebo	30 T1DM (15 adolescents and 15 young adults) 24 h intervention	The primary outcome was TIR. Secondary objectives: average and SD of CGMS readings, TAR, TBR, insulin dose reduction, and urinary glucose excretion	Significant increase in TIR (68 ± 6% Dapagliflozin vs. 50 ± 13% in placebo) and decrease in variability and nocturnal hyperglycemia	No increase	22% reduction in TDD for Dapa	No DKA, but average ketone levels were increased in Empa group vs. placebo (0.29 vs. 0.16 mmol/L, *p* < 0.001)
Haidar et al., 2021 [158]	Empagliflozin 25 mg vs. placebo	30 T1DM adults 9–14 h intervention	To assess whether adding Empagliflozin to HCL could reduce the need for carbohydrate counting in T1DM without worsening glucose control	Similar TIR (68 ± 16% and 70 ± 23%) and CV (31 ± 7% and 30 ± 9%) for meal announcement with Empa vs. carb-counting with no Empa	No difference between groups	Insulin delivery was lower with Empa and carb-counting vs. carb-counting strategy with no Empa (23.7 ± 13.1 vs. 26.2 ± 15.5 U; *p* = 0.054)	Ketone levels were higher in the Empa group (0.22 ± 0.18 mmol/L) than in the placebo group. (0.13 ± 0.11 mmol/L, *p* < 0.001)
Garcia-Tirado et al., 2022 [171]	Empagliflozin 5 mg vs. placebo	35 T1DM adults 2–4 weeks intervention	The primary endpoint was TIR during daytime while on AID. Secondary endpoints: 24/7 TBR, 24/7 average glucose, CV during the day, and risks for hypoglycemia and hyperglycemia during the day	On AID, Empa vs. placebo had higher daytime TIR 81% vs. 71%, *p* = 0.04. On PLGS, daytime TIR was 80% versus 63%, *p* < 0.01.	There is no increase with Empa. Less time in hypo with AID vs. PLGS	TDD was lower both on PLGS (25% and 40%) and AID (24% and 35.5%) with Empa vs. no Empa 24/7 and overnight	One episode of DKA with AID site occlusion contributing. Mild ketosis (>0.6 mmol/L) more frequent on Empa arms
Pasqua et al., 2023 [156]	Empagliflozin 2.5 mg vs. Empagliflozin 5 mg vs. placebo	24 T1DM adults 3 × 14 days intervention	The primary outcome was TIR. Secondary: mean glucose level, SD of glucose levels, CV %, the proportion of participants with TIR > 70%, and TAR, TBR, total daily insulin doses, and mean daily morning ketone levels.	TIR was 70.2 ± 8.0% for 5 mg Empa, 71.6 ± 9.7% for 2.5 mg Empa, and 59.0 ± 9.0% for placebo (*p* < 0.0001)	No difference	TDD were reduced by Empa compared to placebo: 5.2 ± 7.7 u/day with 2.5 mg Empa (*p* = 0.025); and 6.3 ± 4.4 u/day with 5 mg Empa; (*p* < 0.001)	No DKA Events of ketosis > 0.6 mmol/L were few, with slightly more with 5 mg Empagliflozin
Haidar et al., 2022 [178]	Empagliflozin 25 mg	27 T1DM adults 4 weeks intervention	The primary outcome was TIR over 4 weeks.	TIR was 75.5 ± 8.8%, 68.2 ± 9.1%, 69.3 ± 10.7%, and 57.9 ± 13.2% with HCL plus Empa, HCL plus placebo, SAP plus Empa, and SAP plus placebo.	No severe Hypo	Empa reduced TDD compared to placebo both with HCL (*p* = 0.0003) and SAP therapy (*p* = 0.0003) by an identical value of −5.3 U/d (−8.4 to −2.1), RR = 9%.	No DKA Empa increased mean fasting ketone levels.

Abbreviations: TIR—Time in Range (70–180 mg/dL), DKA—diabetic ketoacidosis, TDD—total daily dose, T1DM -Type 1 Diabetes Mellitus, b.i.d.—twice daily, U/d—units per day, SD—standard deviation, TAR—Time above range (>180 mg/dL), TBR—Time below range (<70 mg/dL), CV—coefficient of variation, SAP—sensor-augmented pump, AID—automated insulin delivery system, PLGS—predictive low glucose system, HCL—Hybrid Closed-Loop system, Dapa—Dapagliflozin, Empa−Empagliflozin.

**Table 4 ijms-25-01972-t004:** Reported cases of DKA in patients treated simultaneously with SGLT-2i and HCL system—insulin response of HCL before the event.

Case Report	Treatment	Time to DKA	Glycemia on Admission	Total Daily Dose	Basal Rate	Meal Boluses	Autocorrection
Visser et al. [177]	Empagliflozin 12.5 mg starts on Insulin therapy using Medtronic 780G	2 weeks	353 mg/dL	49% reduction	Reduced	13.6–42% reduction Safe meal bolus frequent	Reduced
Singh et al. [179]	Insulin infusion using Medtronic 670G starts on Empagliflozin therapy	2 weeks in auto-mode	519 mg/dL	15% reduction in auto vs. manual mode 60% reduction Empa vs. no Empa	30% reduced in auto-mode	Similar	No data

## 4. Discussion

Automated insulin delivery systems have significantly improved glycemic control and quality of life for people with T1DM. However, there is one category of patients in whom glycemic or weight control is particularly difficult to achieve. Using SGLT-2i as an add-on to AID therapy might be helpful in this category. The association improved glycemic control, decreased basal and postprandial blood glucose [180], reduced daytime mean glycemia [158], and enhanced glycemic variability. TIR was increased to 68–81% without excess time spent in hypoglycemia. The total daily insulin dose was reduced when SGLT-2i was used with percentages between 10 and 24% 24/7 and up to 40% during nighttime [171,179,181]. Basal and/or bolus insulin doses decreased as a response of the HCL algorithm to glycemic values, which raises the question of whether this might be an underlying mechanism that might increase ketone production and the risk of ketoacidosis.

Although weight control is not one of the main concerns for persons with T1DM, nowadays, the prevalence of overweight and obesity is increasing in this group of patients due to sedentary lifestyle, increased caloric intake, and raised insulin doses. SGLT-2i is one of the molecules that significantly benefit T1DM by up to 5% weight reduction [182,183]. It is one of the main reasons why the off-label use of these compounds by people with T1DM is on a permanent increase. Sotagliflozin and Dapagliflozin’s initial approval for adjunct therapy in patients with T1DM with BMI > 27 kg/m^2^ poorly controlled by conventional treatment T1DM was based on the weight control benefit. A recently published analysis shows that in moderate to long-term therapy (24–52 weeks) of T1DM patients, Insulin+SGLT-2i co-treatment was associated with genital infection (GI) risk. Moreover, Insulin+Sotagliflozin (dual SGLT-2/1 inhibitor) associated therapy was linked with DKA and GI risks [184].

Data on the effect of SGLT-2is in T1DM patients using AID systems are still insufficient as the reported studies had short-term duration, and neither included weight management as one of the endpoints.

The risk of serious adverse events remains the primary concern because SGLT-2i, in conjunction with AID systems, increases ketone levels [182,183,185,186]. Although most DKA-reported cases are associated with pump malfunction issues (such as catheter site occlusion), the concern remains. SGLT-2i increases glucose urinary excretion and diminishes glycemia and insulin delivery by the AID system in conjunction with increased lipolysis, glucagon secretion, and gluconeogenesis, contributing to increased ketone production. The decrease in basal rates frequently seen when SGLT-2is were added to closed-loop systems (CL) [177] and the safe meal boluses mode [179] that activates when glycemia is not significantly increasing may augment the risk of ketoacidosis. Patients should be carefully trained to prevent DKA when using the combination of SGLT-2is and CL systems. Potentially beneficial approaches are adequate patient selection, frequent ketone monitoring protocol, increased vigilance when operating the pump’s auto mode, and effective prevention of stressful situations such as infections, traumas, etc. Ketone monitoring could use a continuous ketone monitor device. Its data could be integrated into an AID algorithm, increasing insulin requirements to reduce ketone levels. However, no details regarding the development of the AID algorithms according to SGLT-2is adjuvant therapy were found in the accessed databases.

A recently published study [187] included data from 250 people living with T1DM receiving Dapagliflozin as an add-on therapy to insulin. The authors reported five DKA events in patients with a 12-month follow-up: two events related to insulin pump malfunction, two events related to concomitant illnesses, and one event connected to insulin dose omission. Interestingly, in the first six months after initiation of Dapagliflozin, DKA events were more frequent among insulin pump users than among MDI-treated patients. No deaths or persistent sequelae due to DKA were reported. No severe hypoglycemia episodes were noted. Significant mean body weight, HbA1c, and total daily insulin dose reductions were observed 12 months after the Dapagliflozin prescription. Substantial improvements in TIR (+9.3%), TAR (−7.2%), TBR (−2.5%), and coefficient of variation (−5.1%) were also observed in the subgroup of patients with available CGMS data. Finally, an improvement in the urinary albumin-to-creatinine ratio (UACR) was found among participants with UACR ≥ 30 mg/g at baseline (median decrease of 99 mg/g in UACR).

Potential long-term benefits of SGLT-2is use, such as decreasing cardiovascular risk [188], HF hospitalization [55,185,189,190], and CKD progression in AID-treated T1DM, can only be supposed as an extrapolation from T1DM and T2DM studies. Long-term randomized trials with cardiovascular endpoints [179] are still missing. Moreover, the adverse events of associating SGLT-2is with AID systems in T1DM patients lead to more attention requested when recommending a therapy with potentially harmful effects.

Increased vigilance from the healthcare provider should be present all the time in case of SGLT-2is add-on, particularly when the patient has poor compliance, does not follow ketone protocol, has a low-carb diet or a low daily insulin dose, or has a recurrent infection or a chronic disease with frequent acute episodes. Therefore, association therapy could show efficacy in improving glycemic controls as adjunctive treatment in T1D, in addition to insulin [191]. A recent study of 26 patients for 12 weeks investigated whether Dapagliflozin and Liraglutide, in addition to an already established regimen of insulin and liraglutide, significantly improved glycemia and body weight. In the triple therapy group, HbA1c decreases by 0.66% versus the placebo group, with an unchanged incidence of hypoglycemia and a body weight decrease (−1.9 kg versus placebo).

Providing the best insulin treatment for the patient with T1D is essential and should prevail over any other treatment option. AID systems have been proven to generate better glycemic control with lower hypoglycemia and higher TIR; therefore, they should be recommended to all patients using such a device. Add-on therapies such as SGLT-2is should be carefully considered as they might induce severe complications. As an overview, the profile of suitable T1DM patients for SGLT-2is associated with AID therapy includes overweighting/obesity (BMI > 27 kg/m^2^), insulin resistance (>0.5 Units/kg), metabolic syndrome, hyperglycemia despite optimized insulin therapy, desire for blood pressure control along with achieving euglycemia, albuminuria, and diabetic nephropathy, and difficulties in reaching TIR > 70%. Contrarywise, it should be avoided in patients of extreme ages (<18 or >75 years old), uncompliant with ketone protocol or with deficits in diabetes self-management, recurrent DKA and other ketosis events, ketogenic or very low caloric diet, BMI ≤ 18.5 kg/m^2^, total dose of insulin received < 0.3 U/kg/day, and recurrent serious infections or acute illness (osteomyelitis, cellulitis, urosepsis, foot ulcers).

## 5. Conclusions and Perspectives

The present review has shown the valuable properties of sodium-glucose cotransporter-2 inhibitors, which makes them a promising therapeutic option in T1DM patients using insulin pumps. SGLTi could optimize AIDs and efficiently add the benefits of weight loss, blood pressure control, and renoprotection that AID alone cannot ensure. SGLT-2is could improve critical diabetic parameters, including glycemic control and weight management. Their main adverse effect is DKA, which is more frequent in T1DM AID users than MDI-treated patients. Continuous verification of AID device parameters and rigorous patient selection could minimize the inherent risks, increasing the combined treatment safety. Thus, a suitable profile of T1DM patients was designed; incompatible issues were also reported.

For the safe use of SGLT-2 inhibitors associated with AID systems in T1DM patients, the mathematical algorithms should be adjusted for this add-on treatment, and a ketone monitoring device with DKA alerts should be integrated into the insulin delivery system.

Further studies can evaluate the effects of SGLT-2is in combination with another drug from another class to determine whether this approach would yield better outcomes, evaluating the DKA risk in patients undergoing triple therapy.

## Figures and Tables

**Figure 1 ijms-25-01972-f001:**
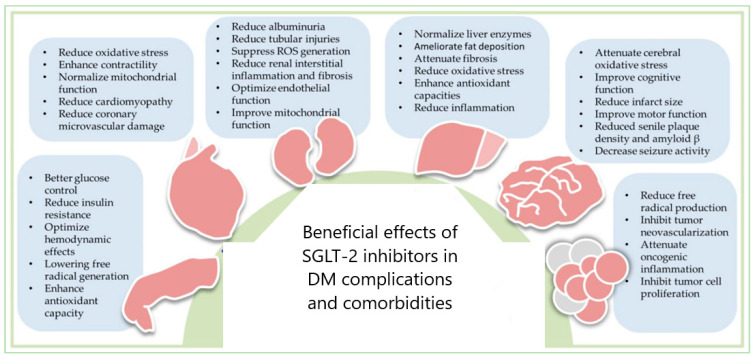
SGLT-2 inhibitors’ therapeutic potential in DM complications and comorbidities: cardiovascular diseases, nephropathy, liver diseases, neural disorders, and cancers—reproduction with permission from [127].

## Data Availability

Data are contained within the article.

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
