# Peer review of "Risks and Benefits of SGLT-2 Inhibitors for Type 1 Diabetes Patients Using Automated Insulin Delivery Systems—A Literature Review"

_ijms, 2024, doi:10.3390/ijms25041972_

Round 1

Reviewer 1 Report

Comments and Suggestions for Authors

Summary: The manuscript titled “Risk and Benefits of SGLT-2 Inhibitors for Type 1 Diabetes Patients Using Automated Insulin Delivery Systems – A Minireview” is a review of the current literature on the risks and benefits of using SGLT-2 inhibitors for patients with T1D which specifically use automated insulin delivery system. The authors indicate in the title that this manuscript is “A Minireview”. However, the written content does not align with that statement. Instead, it aims to provide a rather thorough and broad overview of every aspect of SGLT-2 inhibitors. Therefore, either the title should be changed, or the content of the review should be modified to reflect the narrow focus of the title.

The review provides in-depth information about the above areas with up-to-date, appropriate citations. The manuscript appears somewhat disorganized and lacks the primary focus mentioned in the title. There are seven figures and 2 tables, which is too many for a review paper designated as a minireview. Below are several more specific comments and suggestions for improvement:

1.     Abstract – The abstract needs to be improved in a way that provides a short summary of the topic being reviewed and focuses the readers’ attention on the importance of writing this review. Also, the English language needs improvement. For example, on row 40, the authors state, “It analyzes the mechanisms….”. This is inaccurate as review papers can not analyze mechanisms; they simply critically review the work of others.

2.     Introduction – needs re-writing to introduce the readers to the review's main topic, which is the usage of SGLT-2 inhibitors for patients with T1D using the automated insulin pump. I recommend providing a brief overview of the different mechanisms for glucose uptake in other cells, then focusing on the SGLT-2 and how and why they differ from the GLUT proteins. Then, it transitions to the history of using SGLT-2 inhibitors for patients with diabetes and finally moves to their use in T1D patients. Figures 1 and 2 are unnecessary, especially Figure 2, which represents a basic textbook mechanism that is widely known, and the figure itself does not add any value or novelty to this paper.

3.     Section 2, “SGLT-2 Inhibitors in DM Therapy,” needs to be developed and written appropriately for publication. Currently, it looks like a PowerPoint presentation with a list of bullet points and Smart-Art figures (Figures 3 and 4) that are not only confusing because there is no logic of the colors used or any legend added but also unnecessary. I recommend removing the figures, re-writing this entire section, and combining it with section 3 to include a narrative summary and critical discussion of (1) benefits, (2)adverse effects, and (3) treatment of complications. Table 1 provides a good overview and is appropriately used in this section. The information in Figure 5 is out of the scope of this review. Therefore, I suggest removing it and only providing a summary of the mechanisms underlying the anticancer properties of these drugs, which are still very much in their “exploratory phase” and highly speculative.

4.     Sections 4 and 5 provide the central body of the manuscript that is the focus of this review. Tables 2 and 3 are well-organized and provide valuable information. Figure 6, similar to the other PowerPoint Smart-Art figures, does not add value to the manuscript and is more appropriate for a presentation format than a paper. I suggest removing it and summarizing the content the authors are trying to visualize in the paper's main text.

5.     Discussion – the first paragraph describing the role of oxidative stress in DM and the associated Figure 7 seems entirely out of place at the beginning of this section. I am not sure why they are here, and I suggest removing them entirely from the discussion section as they do not align in any way with the main focus of the paper.

Comments on the Quality of English Language

See main comments.

Author Response

Dear Reviewer 1,

Thank you so much for your excellent comments and suggestions, which, remarkably, the quality of our MS.

We responded point by point to your comments, and we would be grateful to know that we succeeded. 

The MS was rechecked and corrected regarding the English language using performant English software.

Reviewer 2 Report

Comments and Suggestions for Authors This  "mini review" analyzes the impact of SGTL-2 inhibitors on T1DM patients treated
with closed systems on the market nowadays (mainly Medtronic 780 G and Control IQ i.e. T slim and
DexCom G6).
The article is well written and original. However it really starts page 9 (out of 26 ). ..
I should recommend to shorten the first part and focus on:

1- Even if the 5 trials are different is a meta-analysis (GRADE method ) possible ? if so you should do it.
2-Devellop on impact of iSGLT2 on algorithms (PID and MPC)
Comments on the Quality of English Language

None

Author Response

Dear Reviewer 1,

Thank you so much for your excellent comments and suggestions, which, remarkably, the quality of our MS.

We responded point by point to your comments and would be grateful to know that we succeeded. 

Reviewer 3 Report

Comments and Suggestions for Authors

Authors reviewed the potential role of sglt2i as add-on therapy with automatic insulin delivery systems n t1dm patients. The work gathers and suggests an interesting area of research. However, many and important issues must be corrected. Overall, they include an inappropriate introduction with too many physiological concepts. Also, along the work, many redundant text and figures reduce paper comprehension. Some ambiguous data should be also specified. Acronyms must be deciphered at the first time of use and in figure legends. Also, paragraphs must be joined to avoid too much fragmentation.

Abstract: decrease in variability? of what?

Introduction and Results: page 5-8. Authors must avoid schematic writing. It looks they copy and paste from AI. many items should be detailed. In 2.1.1, some potential mechanisms of weight loss are not directly connected such as mTorc1 and ampk activation or polarization of macrophages. Others need more explanation. Figure 3A is redundant with text. In 2.1.2, “potential vascular effects” is ambiguous, as metabolic pathway in 2.1.3. In 2.2.3, there is no apparent reason to include the two last bullets. Similar with the first two bullets in 2.2.4. Figure 4A is not necessary.

Section 3, page 7, where are the emerging complications of diabetes?

Figure 5 gathers data that have not been mentioned in the manuscript. Production of ketone bodies may show a dual face. They can be rapidly used by heart and brain as energetic substrates, avoiding accumulation of fatty acid- or glucose-metabolites. This should be considered in the work. Also, the physiological and pathological ranges of each ketone body must be added.

Figure 6c is redundant with text.

Discussion: it is soundless. The first paragraph is unrelated with results. Key questions must be considered: is there any difference between using sglt2i in t1dm patients with or without automatic insulin devices? Do the sglt21 induce more beneficial or negative effects in patients with these devices compared with those without them?

Table 3 is redundant with text.

Conclusion: second paragraph is repeated with the first

Comments on the Quality of English Language

In general, it is ok

Author Response

Dear Reviewer 3,

Thank you so much for your excellent comments and suggestions, which, remarkably, the quality of our MS.

We responded point by point to your comments and would be grateful to know that we succeeded. 

The MS was rechecked and corrected regarding the English language using performant English software.

Round 2

Reviewer 1 Report

Comments and Suggestions for Authors

The review manuscript is significantly improved compared to the original version. I don't have any additional comments or concerns.

Comments on the Quality of English Language

It looks good, but a minor spell check and grammar may be beneficial.

Reviewer 2 Report

Comments and Suggestions for Authors

None

Comments on the Quality of English Language

OK to me

Reviewer 3 Report

Comments and Suggestions for Authors

Authors somehow answered the required items